# 'Sequencing Economics' on the ICT Industry Agglomeration for Economic Integration

**Akifumi Kuchiki**

Department of Liberal Arts, The Open University of Japan, Chiba 261-8586, Japan; kuchiki.akifumi@nihon-u.ac.jp

**Abstract:** In this paper, we seek to establish 'sequencing economics' in an architectural theory on agglomerations that are comprised of various segments, such as infrastructure, institutions and human resources. The sequencing of such segments is based on a causal chain, with the notion of 'economies of sequence' regarded as a tool with which to efficiently sequence the segment construction, as defined by Granger causality relationships. The use of 'new economic geography' for cases in which such economies of sequence were applied to the information and communication technology (ICT) industry, the paper concludes that as the starting conditions for the sequencing of the segments of the agglomeration, the value of the share of skilled workers exceeds the threshold value at which the symmetric equilibrium shifts to an agglomeration equilibrium. The results of Granger causality testing identified that an increase in research expenses Granger-causes an increase in the number of patents, and an increase in the number of patents Granger-causes an increase in value added. Based on our results, we conclude that when sequencing the segments of an agglomeration in the ICT industry, the development and invitation of researchers is a precondition, and that the procurement of research funds for patent development precedes any increase in patents. Subsequently, the procurement of funds is necessary for the development of products based on the patents.

**Keywords:** sequencing economics; economies of sequence; architecture theory; researcher; funding support; patent development; product development; the ICT agglomeration

**JEL Classification:** O18; R12



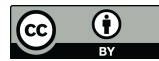

## 1. Introduction

The target of the ASEAN Economic Community 2015 is to enhance infrastructure and communication connectivity. The One Belt One Road Initiative is a Chinese project that focuses on improving connectivity. Strengthening physical, intuitional, and people-to-people connectivity promotes regional economic integration by reducing transport costs.

Transport costs in new economic geography include the opportunity costs of the days required for transportation, tariffs, non-tariff barriers, and differences in language and cultural customs (Sato et al. 2011). The reduction in transport costs is a necessary condition for industrial agglomeration, resulting in economic integration. According to Fujita and Thisse (2003), "the main effect of regional integration is likely to be an increase in economic efficiency within the spatial economy". We herein focus on regional integration from an agglomeration perspective.

'Sequencing economics' is used to specify and sequence the segments of an agglomeration from the perspective of 'economies of sequence'. The concept of economies of sequence can be defined as the selection of any two segments from among the entire group of segments of an industrial agglomeration and the sequencing of the segments toward the efficient building of the agglomeration. Granger causality tests examine whether the facts of 'economies of sequence' are significant or not.

According to Kuchiki (2019), theories on agglomeration can be classified into theories of location, architecture, and geographic management (Table 1). First, (i) the new economic

geography in location theory revolves around the conditions of spatial concentration, or agglomeration. Fujita et al. (1999) developed a spatial version of the Dixit–Stiglitz model of monopolistic competition to study where economic activity occurs and why. Second, (ii) sequencing economics provides an architectural theory on agglomerations. Third, Porter (1990) established (iii) a diamond model with which to examine a set of location advantages that contribute to the competitiveness of a region in terms of geographic management theory.

**Table 1.** Three theories on agglomeration.

| Theory | (i) Location | (ii) Architecture | (iii) Geographical Management |
|---|---|---|---|
| | Fujita et al. (1999) | Kuchiki (2019) | Porter (1990) |
| Model | (i) New Economic Geography | (ii) Sequencing Economics | (iii) Diamond Model |
| Characteristics | To show how a two-region can become a core-periphery pattern | Sequential processes in efficiently building the segments of an agglomeration | Finding the factors of "competitive advantages" of a region |
| Key factors | The Dixit–Stiglitz model, economies of scale, and transport costs | Integration of 'economies of sequence' between two segments | 1. Demand conditions; 2. Factor conditions; 3. Firm strategy, structure, rivalry; 4. Related and supporting industries |

Source: Kuchiki, based on the table by A. Kuchiki and I. Ohno in Kuchiki (2019) and Kuchiki (2006).

Table 1 and Figure 1 clarify the relationships between (i) new economic geography and (ii) sequencing economics among the three theories. Regarding the economies of sequence, many models on spatial economy in the manufacturing industry conclude that the starting conditions that reduce transportation infrastructure costs, such as road and port construction, are the first priority, as shown in Figure 2.

With regard to new economic geography, Krugman (1991) proposed a fundamental model by introducing the mobility of skilled workers. Martin and Ottaviano (2001) developed a dynamic version of this fundamental model by imposing an assumption that research and development is the source of growth. Hirose (2008) analyzed the case of migration and agglomeration using a dynamic version. Fujita and Thisse (2003) assumed that skilled workers are mobile and that patents for new products can be transferred without cost between regions. As shown in Figure 1, their model can be applied to a model for a two-region conglomeration across two counties to drive the conditions of an information and communication technology (ICT) agglomeration.

With regard to sequencing economics, Kuchiki (2020a) analyzed manufacturing industry agglomerations, including the electric/electronics industry and the automobile industry, as shown in Figure 2. It showed successful agglomeration policies in East Asia from the 1980s. Kuchiki (2020b) also applied sequencing economics to tourism industry agglomerations.

With regard to ICT agglomerations, Han et al. (2019a) and Monti and Ponci (2017) published Special Issues on urban agglomeration and ICT, respectively. Chen et al. (2019) and Han et al. (2019b) also analyzed high-tech industry agglomerations based on green economy efficiency and industrial agglomerations based on land use efficiency, respectively.

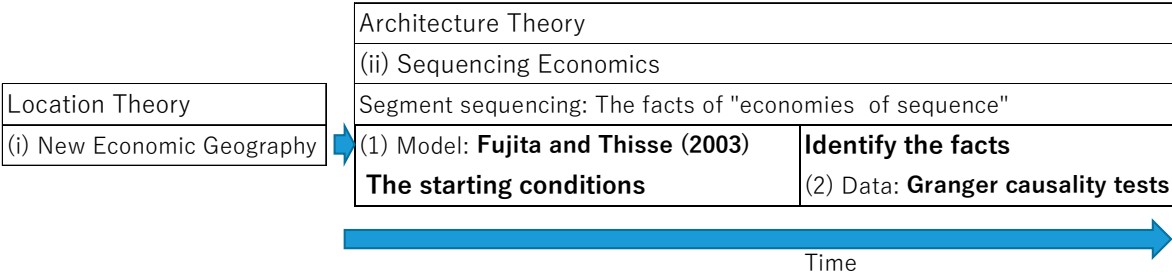

**Figure 1.** The relationship between new economic geography and sequencing economics. Source: Author.

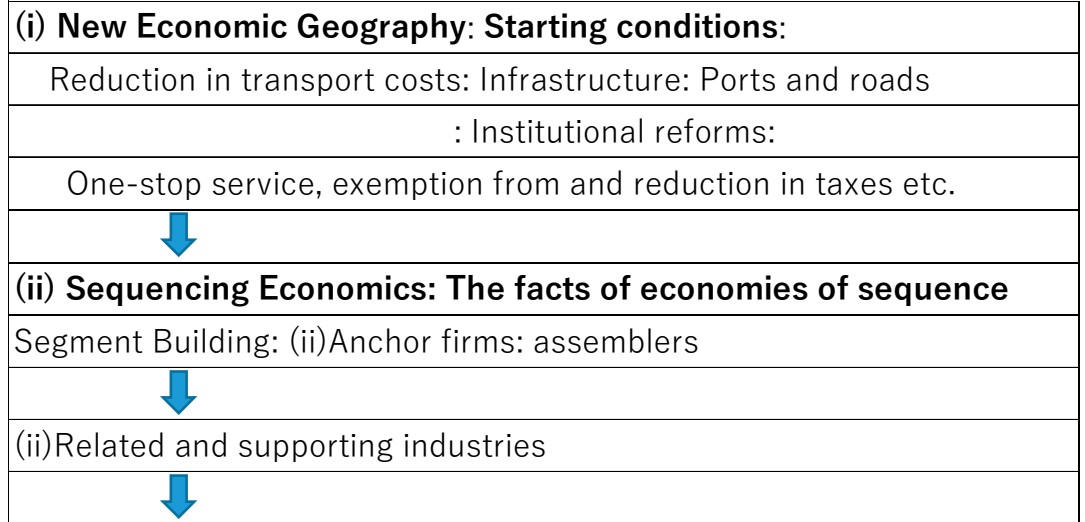

**Figure 2.** The starting conditions for the manufacturing industry derived by new economic geography. Source: Author.

However, 'sequencing economics' in ICT industry agglomerations is yet to be analyzed. Table 2 shows the segments of an ICT industry, including researchers and funding. The building of the segments of an agglomeration can be started once they are sequenced efficiently. The economics seeks to reveal the 'economies of sequence'—i.e., how the efficient construction of the segments of an agglomeration should be sequenced. Thus, this paper applies the model of Fujita and Thisse (2003) to the case of a two-region model across two countries to drive the starting conditions for constructing agglomerations in the ICT industry.

The purpose of this paper is to find the facts of 'economies of sequence' in an ICT industry agglomeration while trying to establish 'sequencing economics' on the agglomeration in architecture theory. The Zhongguancun Science Park in Beijing, China (ZSP), is a case study of an ICT industry agglomeration, which has been taken as a reference. The paper focuses on sequencing economics in architecture theory by taking 'new economic geography' into consideration. The model of Fujita and Thisse (2003) provides the starting conditions of the sequencing of the segments of the ICT industry agglomeration, as shown in Figure 1. Integrating the results of the three studies on economies of sequence proposes practical prescriptions for segment sequencing. In short, sequencing economics analyzes the segment building of an ICT agglomeration from the three sequencing perspectives of a case study, a model of new economic geography, and Granger causality testing for practical application to a regional development strategy.

**Table 2.** The segments of information and communication technology (ICT) agglomerations.

| Segment | |
|---|---|
| **Human resources** | (1) Development and invitation |
| **Persons in charge of commercialization (Granger test)** | |
| Infrastructure | (2) Parks, sub-parks |
| | Railway |
| | Airport |
| | Port |
| | Communication |
| | Water |
| | Electricity |
| | Roads |
| **Institutions** | (3) RIS, UIGLs |
| | (4) Funding support |
| | **Funds (Granger test)** |
| | Laws and regulations |
| Living conditions | Housing and entertainment |
| | Hospitals and schools |
| | The environment |

Source: Author.

The paper concludes that as the starting condition, the value of the share of skilled workers exceeds the threshold value at which the symmetric equilibrium shifts to an agglomeration equilibrium. Granger causality tests identified that first, an increase in research expenses Granger-causes an increase in the number of patents, and second, an increase in the number of patents Granger-causes an increase in the amount of value added. As shown in Figures 1 and 2, these facts of 'economies of sequence' are the necessary conditions for agglomeration.

The paper is structured as follows. Section 2 illustrates ZSP, a case study on sequencing economics, as a reference. In Section 3, the conclusions of Fujita and Thisse (2003) provide the starting conditions for the sequencing of the segments of an ICT agglomeration. In Section 4, Granger causality tests identify the facts of 'economies of sequence'. Section 5 is the summary and conclusions, providing an integration of the results of Sections 2–4 above.

## 2. Sequencing Economics in the ICT Industry

### 2.1. The First Stage (Agglomeration) and the Second Stage (Innovation)

Based on the definition of an industrial cluster provided by Fujita (2003), who states that a cluster is produced when an agglomeration is a means of stimulating innovative activities, Kuchiki (2007) introduced the sequential process of agglomeration segment construction and divided it into two stages: the first stage being agglomeration, and the second stage being innovation (Figure 1).

Figure 2 presents the two stages in the sequencing economics of the manufacturing industry, as summarized by Kuchiki (2020a). The left side shows the flow of the first stage of agglomeration and the right side is that of innovation. In Stage 1, the location theory leads us to accurate decision-making regarding how we sequence the segments of an agglomeration for their efficient construction. The segments related to Stage 2 are the four factors of demand conditions, factor conditions, firm strategy, and supporting industries.

The segments of an ICT agglomeration play key roles in explaining the innovations in Hershberg et al. (2007) and Brimble and Doner (2007). The sequencing economics of the ICT industry agglomeration is illustrated in Figure 3, which demonstrates the construction of ZSP in Beijing, China. The existence of universities and research institutes is a precondition to building the segments of an ICT agglomeration. As shown in Table 2, an ICT agglomeration consists of five main segments: (1) the development and invitation of researchers and managers on human resource development; (2) hard infrastructure,

including parks and sub-parks; (3) institutional reforms, including university–industry–government linkages (UIGLs) and a regional innovation system (RIS); (4) funding support, and (5) a living environment. Hershberg et al. (2007) reviewed the literature on university–industry linkages (UILs) and RISs.

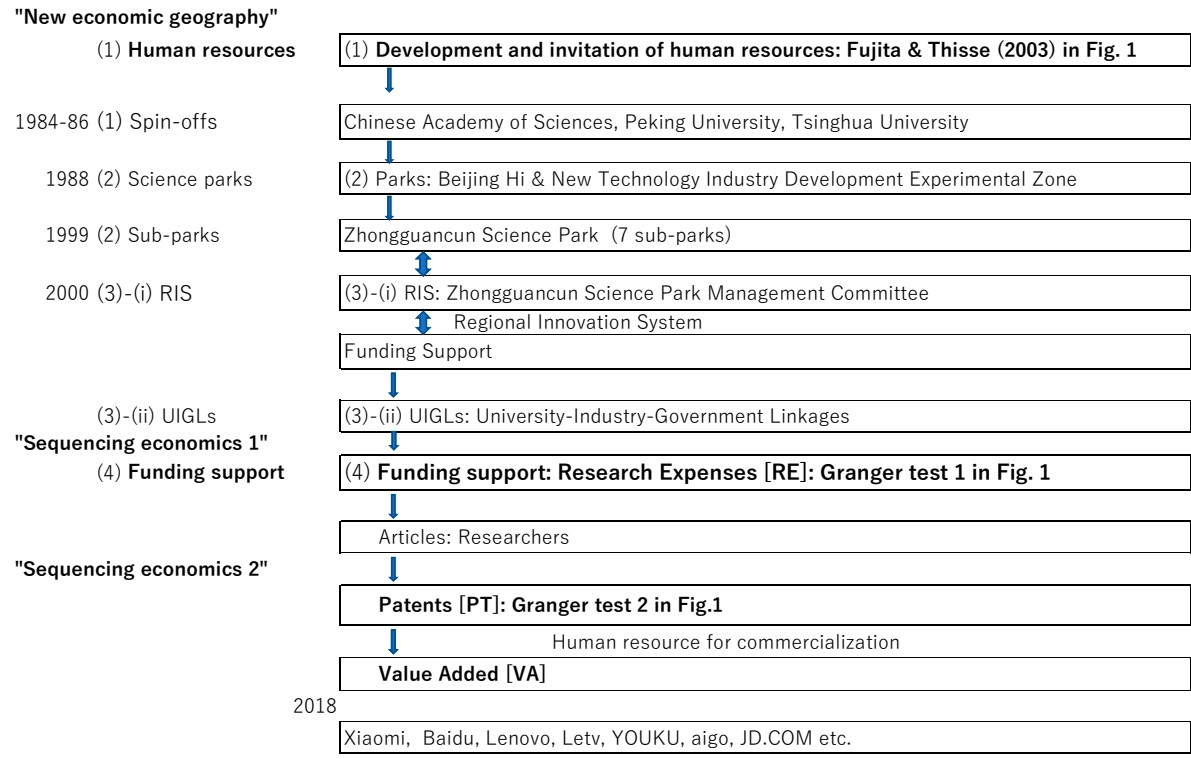

**Figure 3.** Sequencing of the segments of the ICT agglomeration at the Zhongguancun Science Park in Beijing. Source: Author.

### 2.2. Economies of Sequence

We will explain the concept of "economies of sequence" and show two examples of segment construction sequencing in an ICT agglomeration, as Kuchiki (2019) stated. Assume that there are n periods. Two sequences (S and T) build a segment. Suppose that an agglomeration is comprised of n segments $\{s_1, s_2, s_3, \ldots, s_n\}$, say {the development of professionals, a science park, funding support, . . . , patents} in the manufacturing sector model.

The difference between S and T is the sequence of $s_1$ and $s_2$. In S, the segment building sequence is in the order $s_1$ and $s_2$. In T, it is the sequence of $s_2$ and $s_1$. That is,

$S = \{s_1, s_2, s_3, \ldots, s_n\}$ = {the development of professionals, a science park, funding support, . . . , patents}, and

$T = \{s_2, s_1, s_3, \ldots, s_n\}$ = {the development of professionals, a science park, funding support, . . . , patents}.

Putting the costs of $S = \{s_1, s_2, s_3, \ldots, s_n\}$,

$C_s = f(\{s_1, s_2, s_3, \ldots, s_n\})$, and

$C_T = h(\{s_2, s_1, s_3, \ldots, s_n\})$.

The costs of $C_T$ are much higher than those of $C_s$ because of the mistake of sequencing the order of $s_1$ and $s_2$.

This section defines the concept of "economies of sequence" using the Granger causality test. The sequence economy is held in the case that, in the Granger causality test between $s_1$ and $s_2$, a null hypothesis that $s_1$ does not Granger-cause $s_2$ is rejected.

Consider the two variables of x and y. Equation (1) is an autoregressive model of y and Equation (2) is an autoregressive model of x and y. Granger causality holds if the prediction error value of Equation (2) is smaller than that of Equation (1).

Equations (1) and (2) without a drift term are

$$y(t) = b_{11}\, y(t-1) + b_{12}\, y(t-2) + \ldots + b_{1n}\, y(t-m) + e_1, \tag{1}$$

$$y(t) = b_{21}\, y(t-1) + b_{22}\, y(t-2) + \ldots + b_{2n}\, y(t-m) + c_1\, x(t-1) + c_2\, x(t-2) + \ldots + c_2\, x(t-m) + e_2, \tag{2}$$

where the lags of x and y are m (m = 1, 2, 3, 4, or 5, in our models) and $e_i$ (i = 1, 2) is an error term. Equation (1) uses OLS (Ordinary Least Squares) regression analysis, and its sum of squared residuals is notated as $SSR_o$; Equation (2) is uses OLS regression analysis, and its sum of squared residuals is notated as $SSR_1$.

The F-test is defined as

$$F = ((SSR_o - SSR_1)/m)/(SSR_1/(T-K)), \tag{3}$$

where m and T − K are the numbers of degrees of freedom, T is the sample size, m is the number of lagged x and y terms, and K (=2 m) is the number of parameters estimated in the regression of Equation (2).

### 3. The Key Segments of an ICT Agglomeration at Zhongguancun Science Park (ZSP)

Kuchiki et al. (2017) proposed the sequencing economics of the ICT industry agglomeration policy. Table 2 illustrates the segments of an ICT agglomeration and Figure 3 is a summary of its sequence from the agglomeration of (1) human resources starting from spin-offs at the Zhongguancun Science Park (ZSP).

First, regarding the segment of (1)—(i) the development and (1)—(ii) the invitation of human resources, spin-offs from research institutes and universities were a precondition to start the sequencing economics of the ICT agglomeration policy, mainly from 1984 to 1986. Second, regarding the segment of (2) parks and sub-parks, in 1988, Zhongguancun Electronic Street was renamed the Beijing Hi and New Technology Industry Development Experimental Zone and officially became the Zhongguancun Science Park (ZSP). Third, regarding the segment of (3)—(i) a regional innovation system (RIS) of institutions, the Zhongguancun Science Park Management Committee, as a regional innovation system, functioned effectively, implementing the ICT agglomeration policy in ZSP. Its preferential treatment was effective in inviting human resources to ZSP. Fourth, regarding the segment of (3)—(ii) university–industry–government linkages (UIGLs) of institutions, their examples are shown below. Fifth, regarding the segment of (4) funding support, the regional innovation system (RIS) of the ZSP Management Committee and university–industry–government linkages (UIGLs) played a role in supporting funds. This section explains the five segments above one by one.

### 3.1. The Segment of Human Resources

3.1.1. Human Resource Development

Agglomeration, Stage 1 in the flowchart, was achieved through (1) the development of human resources at the universities and research institutes in Figure 3. Peking University, Tsinghua University, and the Chinese Academy of Sciences (CAS) functioned well in agglomerating two-thirds of all of the Chinese PhDs in Beijing. The Institute of Policy Management (IPM) of the CAS, established in 1985, has contributed to recruiting 62 doctoral students since 1995.

The case (1) of spin-offs in which universities and research institutes, including the Chinese Academy of Sciences (CAS), Peking University, and Tsinghua University, established firms is shown in Table 3. Many of the famous companies in ZSP were founded in 1984–1985.

**Table 3.** Institute/university industry linkages.

| Research Institutes/Universities | Name of Enterprise | Notes |
|---|---|---|
| Institute of Computing Technology, Chinese Academy of Sciences | Lenovo Group Ltd. | Development, manufacture and sales of computer products |
| Software Research Institute of Chinese Academy of Sciences | Red Flag Software Co., Ltd. | Ministry of Information Industry invested a hundred million yuan in 2001 |
| Chinese Academy of Science | China Sciences Group (Holdings) Corporation | Zhong Ke San Huan, Zhong Ke Hope Software, Shanghai China Science, Dayang and CSCA Technology, etc. |
| | China Daheng Group Inc. | Optical components, China's Top 100 Electronics Enterprises |
| Peking University | Peking University Founder Group Corp. | Computer and multimedia products |
| | Beida Jade Bird Group | Software production, mainly software development, system integration, and computer security |
| Tsinghua University | Tsinghua Tongfang Co. Ltd. | Computer products |
| | Tsinghua Unisplendour Co. Ltd. | Environmental protection industry |

Source: Kuchiki, based on Kuchiki (2007).

### 3.1.2. The Invitation of Human Resource: Beijing Residency Permits

Table 4 shows the preferential policies for hi-tech enterprises to introduce human resources to ZSP. Preferential treatment regarding the family register, particularly Beijing residency permits, has been effective in inviting "human resources."

**Table 4.** The segments of human resource and funding.

| **(1) Human Resource** |
|---|
| Persons studying abroad or technical and managerial talents from other provinces or cities: Beijing residency permits. Their children: compulsory education. Newly graduated students from colleges and universities: their entering Beijing. |

| **(2) Funding** |
|---|
| New and innovative technology, risk investment, and guarantee funds. Risk investment institutions: the form of limited liability of partnerships. Beijing Municipality subsidies: a limit of ten million yuan. Subsidies up to 50% of the interest on loans. Small loans to enterprises established by students returning from overseas: a limit of one million yuan. Incubator Fund established within the past three years: a limit of three million yuan. Special funds: senior managerial personnel and technical personnel in software enterprises and IC enterprises. Integrated Circuit Design Enterprise Loans. |

Source: Kuchiki, based on Kuchiki (2007).

In terms of policy on the invitation of human resources, the IPM of the CAS recruited students studying abroad to work in ZSP. The IPM undertook a plan to send back 100 researchers to ZSP and achieved this target.

The model of Fujita and Thisse (2003) provided a new economic geography to derive the starting conditions for the sequencing of the segments of an ICT agglomeration. Fujita and Thisse (2003), in their new economic geography, concluded that both the modern and innovation sectors agglomerated within the same region under the condition of sufficiently low transport costs. The model supposes that two regions are 1 and 2, that production sectors are the traditional sector, the modern sector, and the innovation sector, and that two production factors are low-skilled workers and high-skilled workers. The innovation sector employs high-skilled workers, the total number of skilled workers in the economy is fixed, and each skilled worker can change regions at some positive cost.

The number of unskilled workers is the same in each region over time and each unskilled worker cannot move.

As the starting conditions of the sequencing of the segments of the ICT industry agglomeration, the value of the share of skilled workers exceeds the threshold value at which the symmetric equilibrium shifts to an agglomeration equilibrium. Beijing residency permits contribute to an increase in the share of skilled workers in a region. Under the low transport costs, the policy of raising the share of skilled workers in a region, λ,

$$\text{from } \lambda < \lambda'' \text{ to } \lambda'' \leqq \lambda,$$

results in $M_1 = M$ and $M_2 = 0$. When the transport costs of the modern goods are low, all of the firms in the innovation sector are agglomerated in the core region (see Appendix A).

*3.2. The Segment of Science Parks and Sub-Parks*

Zhongguancun Science Park (ZSP)

The numbers of firms and employees in Zhongguancun Science Park (ZSP) were approximately 9500 and 400,000 in 2002, respectively. In 2006, 39 universities, 75 national engineering research institutes, and 71 nationally important laboratories were located in ZSP. In 2017 alone, nearly 30,000 firms in ZSP reported a total revenue of more than CNY 5 trillion and held 43,000 patents.[1]

The number of sub-parks in ZSP changed from seven in 2005 to 16 in 2017. The main sub-park in 2007 was Haidian Science Park (HSP), which covered 217 square kilometers and managed 20 sub-parks. By 2004, approximately 10,000 firms had moved into HSP, which covers various types of industries, such as electronics, information technology, opt mechatronics, and new materials. The number of researchers was 378,000 in 2006, as shown in Table 5.

**Table 5.** Human resources of the science sub-parks in 2006.

| | Park | District | Surface Area (km²) | Number of Companies | Research Institutes/Universities | Researchers |
|---|---|---|---|---|---|---|
| 1 | Haidian Park | Haidian District | 217 | Over 10,000 | Research institutes: 232 Universities: 73 | Researchers: 378,000 University students: 300,000 |
| 2 | Changping Park | Changping District | 5 | Over 1300 | Research institutes: 114 Universities: 14 | Researchers: 15,000 |
| 3 | Fengtai Park | Fengtai District | 5 | Over 2700 | Research institutes: 60 | Researchers: 70,000 |
| 4 | Yizhuang Park | Inside the Beijing Economic-Technological Development Area | 7.5 | Over 1000 | — | — |
| 5 | The Electronic Zone | Jiuxianqiao, Chaoyang District | 10.5 | Over 440 | Electronic research institute: 4 Electronic universities: 5 | — |
| 6 | Desheng Science and Technology Park | Xicheng District | 6 | 145 | CAS Institutes: 12 | CAS academic members: 44 CAE academic members: 5 TWAS academic members: 5 |
| 7 | Jianxiang Science and Technology Park | Chaoyang District | 4.2 | — | Research institutes: 6 Universities: 8 | — |

Notes: CAS: Chinese Academy of Sciences; CAE: Chinese Academy of Engineering; TWAS: Third-World Academy of Science. Source: Author, based on Zhongguancun Science Park Management Committee, 2006.

---

[1] Administrative Committee of Zhongguancun Science Park. 2018. 2 December. Available online: http://www.chinadaily.com.cn/m/beijing/zhongguancun/2018-02/12/content_35696638.htm (accessed on 17 February 2019).

The firms moving into HSP helped Beijing to build the segments of an industrial agglomeration to attract the research institutes of multinational corporations. The branches of world-leading enterprises, such as Microsoft and IBM[2], are located in Tsinghua Science Park, as shown in Figure 4. Tsinghua Science Park (THSP) and Peking University Science Park are located in HSP. THSP, established in 1994, was a holding company for Tsinghua University. Around 63.3% of ICT companies in THSP belong to the electronic information industry.

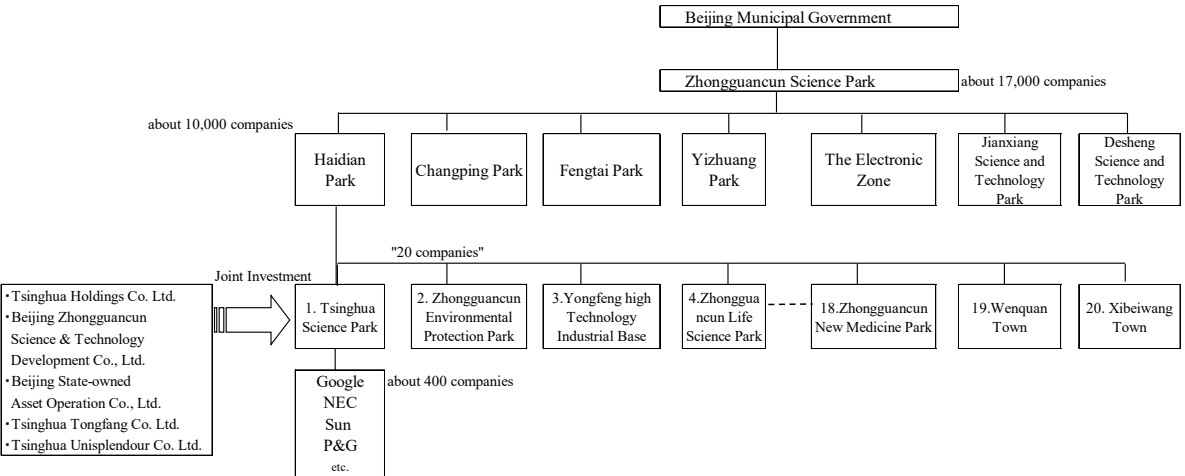

**Figure 4.** The regional innovation system of management structure of the ZSP in 2000. Source: Author, based on Kuchiki (2007).

### 3.3. The Segments of Institutions

#### 3.3.1. The Segment of Regional Innovation System (RIS)

The State Council functions as a regional innovation system (RIS) and supports funding and human resources. In 2000, the State Council authorized the Zhongguancun Science Park Management Committee for ZSP.[3] One of its major roles is to intensify the linkages of firms, universities, and governments.

Figure 4 explains the RIS, and the Beijing Municipal People's Government ranks first at ZSP. The mayor becomes a chairperson of the ZSP Group and the eighteen members of the group comprise the presidents of Tsinghua University, Peking University, and so on. The Zhongguancun Science Park Management Committee substitutes the Beijing Municipal People's Government to manage the Zhongguancun Science Park Group.

The Beijing Municipal People's Government issued a municipal ordinance on ZSP at the end of 2000. Regarding its agglomeration policy, the ZSP ordinance provides (1) development and invitation of human resources and (4) funding support, as shown in Figure 3.

#### 3.3.2. The Segment of University–Industry–Government

Figure 5 illustrates the types of linkages between universities and firms in the agglomeration of firms in the ICT industry in ZSP as follows: first, professors and students at universities 'start up' firms; second, universities and firms implement 'joint research'; third, firms 'fund' universities for producing new products; fourth, firms provide 'employ-

---

ment opportunities' to universities; fifth, the researchers of firms study at universities to undertake a 'Ph.D.'; and sixth, universities sell their 'research findings' to firms.

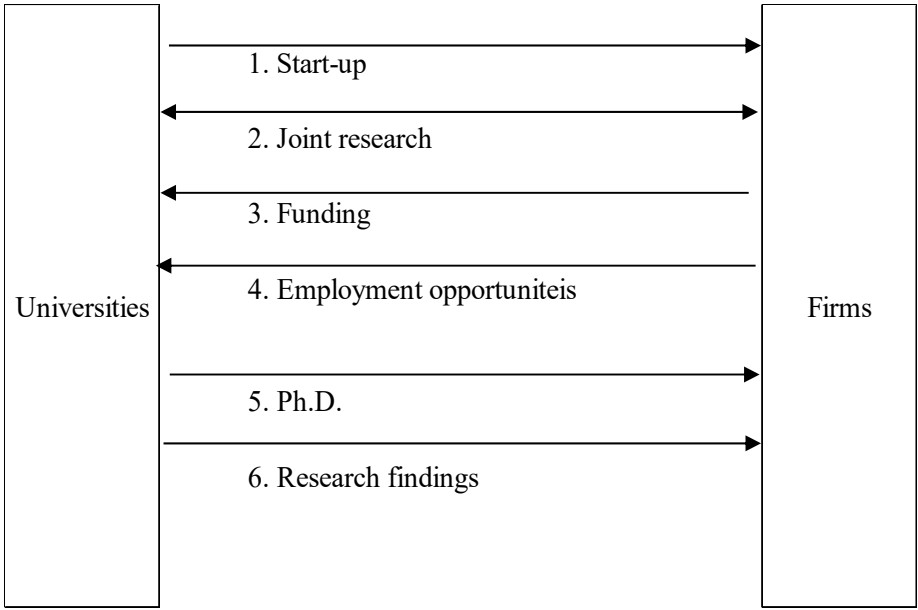

**Figure 5.** The linkages between universities and companies in the ZSP. Source: Author.

### 3.4. The Segment of Funding Support

The "fund" mentioned is meant to support technological innovation by small- and medium-scale enterprises through the provision of funding. The Beijing Municipal People's Government provides professional funds for the ICT industries in ZSP, as shown in Table 4. The interest rate is subsidized by 1.5% for credit on items associated with integrated circuits (IC).

In 2013, ZPS provided five kinds of funding as follows. First, Open Laboratory Subsidy Funds were disbursed from the ZPS Special Development Fund. This money was used for inspection, testing, and technological breakthroughs and to undertake joint major technical innovation and industrialization projects. Second, the ZPS Subsidy Fund for Technical Standards was mainly used to support, free of charge, the three areas, such as technical standards, stipulated under the leadership of hi- and new tech enterprises in ZSP. Third, the Management Method of Support Fund for Financing Guarantees mainly supported enterprises established by oversea students. Fourth, the Zhongguancun Patent Promotion Fund was dedicated to patent promotion, improving enterprises' awareness of patents and to promote effective management. Lastly, enterprises that were listed through the quotation transport system and that received loans from banks could apply for the Support Fund for Equity Pledge Loans to receive a discount loan interest rate.

### 3.5. The Results of the Agglomeration Policy Implemented by Beijing City

In sum, the segments key to activating innovation were human resources, funding, parks and sub-parks, an RIS, and UIGLs. Regarding the trends in the development of the ZSP enterprises in terms of patents from 1996 to 2012, the null hypothesis of Granger causality is that the number of applications does not Granger-cause the number of grants, and the *p*-value of 0.01397 was considered statistically significant at $n = 2$.[4] This implies that fund support is a priority in sequencing the segments of an ICT agglomeration in the second stage of innovation. As shown in Figure 3, the implication is reconfirmed in the next section.

---

4   South China Morning Post. 2018. Mandy Zuo, 5 March. Available online: https://www.scmp.com/author/mandy-zuo (accessed on 30 March 2018).

The Fortune Global 500 list in 2017 makes it clear that the industrial agglomeration policy has been successful. For example, the number of Chinese, Japanese, and U.S. firms listed in the Fortune Global 500 in 2016 numbered 103, 52, and 134, respectively. However, Beijing had the most of any city in the world in 2017, with the numbers for Beijing, Shanghai, and Shenzhen being 56, 6, and 8, respectively. Further, state-owned firms in Beijing numbered 52 out of a total of 56.

In the first decade after the green card scheme was launched in 2004, 7356 documents were issued. In 2016, the country extended eligibility for permanent residency to foreigners working in a wider range of areas. Wang Xi, the Deputy Director of the Beijing Public Security Bureau's Exit and Entry Administration, stated that 662 foreigners filed permanent residency applications in Beijing last year, up from 598 in 2016 and triple the number recorded for 2015.[5]

In 2018, world-renowned firms located in ZSP included Xiaomi, Baidu, Lenovo, Letv, aigo, iSoftStone, and JD, to name a few.[6] The National Cultural and Creative Experiment Zone located in Beijing's Chaoyang District was established in July 2014, after which the number of firms increased from approximately 16,000 in 2014 to 36,625 as of the end of August 2017.

## 4. Granger Causality Tests on the ICT and Computer Industries

This section shows that the segments of both 'funds support' and 'human resources support' in a regional innovation system are a priority when sequencing the segments of an agglomeration. For Stage 2, the innovation stage, we examined whether the sequence of research expenses, the number of articles, the number of patents, and the amount of value added could be identified using Granger causality tests.

Here, the data used for the information communication technology industry on research expenses correspond to ISIC Rev. 4 26 of the manufacture of computers and electronic and optical products. The classification 262 is the manufacture of computers and peripheral equipment and 263 is the manufacture of communication equipment.

Table 6 shows the two following results. First, an increase in research expenses Granger-causes an increase in the number of patents in the ICT industry in Italy, Korea, and Germany. As seen, the null hypothesis of Granger causality is that research expenses do not Granger-cause the number of patents in Italy, and the $p$-value of 0.1033 was considered statistically significant at $n = 1$, or the lag of one year, with regard to the information and communication technology industry. The null hypothesis of Granger causality is that research expenses do not Granger-cause the number of patents in Korea, and the $p$-value of 0.09666 was considered statistically significant at $n = 1$. The null hypothesis of Granger causality is that research expenses do not Granger-cause the number of patents in Germany, and the $p$-value of 0.06848 was considered statistically significant at $n = 4$.

5   Du Yan. 2019. Chinanews.com, Beijing, 11 September. Available online: http://english.iccie.cn/web/static/articles/catalog_ff80808133067bbf01330 fc4f7050016/article_ff8080815becfb08015e78db27ec5107/ff8080815becfb08015e78db27ec5107.html (accessed on 11 September 2019).

6   Wang Hongyan. 2018. Administrative Committee of Zhongguancun Science Park, Tokyo Office. Available online: https://www.vipo.or.jp/u/1803 08_2.pdf (accessed on 11 December 2018).

**Table 6.** The results of Granger causality tests (without a drift term).

| | Industry | Research Expenses [RE] | | | # of Articles [AC] | # of Patents [PT] | Value Added [VA] |
|---|---|---|---|---|---|---|---|
| Italy | ICT Computer | [RE] ⊟ | ⊟ | | ⊟ | (1) 0.1033 [RE] <br> [PT]⊟ | (2) 0.09378 [PT]) |
| Korea | ICT Computer | [RE] ⊟ | ⊟ | | ⊟ | (1) 0.09666 [RE] | (4) 0.0177 [PT] |
| Germany | ICT Computer | [RE] ⊟ | ⊟ | | ⊟ | (4) 0.06848 [RE] | (1) 0.04618 [PT] |
| USA | ICT Computer | [RE] ⊟ | ⊟ | ⊟ | (3) 0.0001304 [RE] | ⊟ (1)0.02972 [AC] | (2) 0.1167 [PT] |
| Japan | ICT Computer | | | | | ⊟ (2)0.03777 [AC] | (2) 0.001552 [PT] |
| China | ICT Computer | [RE] ⊟ | ⊟ | ⊟ | (3) 0.000001894 [RE] | [PT]⊟ | (3) 0.1017 [PT] |
| UK | ICT Computer | | | | | [PT]⊟ <br> [PT]⊟ | (4) 0.02669 [PT] <br> (4) 0.06605 [PT] |
| France | ICT Computer | | | | | [PT]⊟ | (3) 0.004153 [PT] |
| India | ICT Computer | | | | | [PT]⊟ | (3) 0.06326 [PT] |

Note: ( ) are the number of lags; [ ] are causes of Granger causality tests; where RE, AC, and PT mean research expenses, articles, and patent, respectively. The arrow of ⊟ means that the origin of a cause goes to the next cell. Source: Author, using UNWTO. Data extracted from GLOBAL NOTE https://globalnote.jp/ (Accessed on 29 April 2019) and Appendix B.

Second, an increase in the number of patents Granger-causes an increase in the amount of value added by the ICT industry in Korea, Germany, United Kingdom, France, and India. Third, similarly, an increase in the number of patents Granger-causes an increase in the amount of value added by the computer industry in Italy, the U.S., Japan, China, and United Kingdom.

## 5. Summary and Conclusions

This paper sought to apply sequencing economics in the architectural theory of agglomeration to an ICT agglomeration for regional economic integration. 'Economies of sequence' can be defined as the selection of any two segments from among the entire group of segments of an industrial agglomeration and the sequencing of the segments toward the efficient building of the agglomeration, using Granger causality testing.

The paper referenced the agglomeration of the Zhongguancun Science Park in Beijing, China, as an example of the successful sequencing of segments of an ICT industry agglomeration. The main segments consist of (1) the development and invitation of human resources, (2) creation of industrial parks as physical infrastructures, (3) RISs and UIGLs in institutions, and (4) funding support.

The paper consisted of the following three parts: first, the case study of the Zhongguancun Science Park as a reference; second, the model of Fujita and Thisse (2003) in a new economic geography to derive the starting condition for the sequencing of the segments of an ICT agglomeration, that is, the segment building of raising the share of researchers; third, Granger causality testing to identify the facts of 'economies of sequence'. To summarize the major findings, an increase in research expenses was found to Granger-cause an increase in the number of patents, and an increase in the number of patents Granger-causes an increase in the amount of value added. To conclude, the economies of sequence of the segments of an ICT industry agglomeration can be described as follows: a precondition is the development and invitation of researchers, the first priority is the procurement of research funds for patent development, and the second priority is the procurement of funds for the development of products based on the patents.

Six main issues remain to be examined in future. First, this paper used Beijing as a case study for an ICT industry agglomeration and obtained tentative conclusions to apply to other cases in other regions around the world. The number of case studies should, however, be increased to ensure general applicability of the conclusions obtained in the paper. Second, sequencing economics is to be adopted to analyze not only the manufacturing and tourism industry but also other industries, despite this paper focusing only on the ZSP ICT industry agglomeration. Third, with regard to statistical analysis, new data should be explored and new methods of statistical analysis should be developed over and above Granger causality tests. Fourth, the key new segments of an agglomeration need to be identified to clarify substitute data for segments, prove the 'economies of sequence', and establish the 'sequencing economics' to agglomeration policy. Sixth, environmental issues related to agglomeration are a key issue that should be tackled as soon as possible after the COVID-19 pandemic has been controlled. Finally, the feasibility of the proposed methodology is key to completing the building of the segments of an agglomeration. The methodology fits quite well with the analysis of the Chinese case, where there is the coexistence of 'market' and 'indicative planning'. However, we need to examine whether the methodology, specifically the sequencing of phases proposed in Figure 2, can be applied to socio-political and cultural contexts in which the State has the capacity of acting on the provision of resources, such as human resources and infrastructure, but faces limitations on the promotion of business activities.

**Funding:** This work was supported by Japan Society for the Promotion of Science (JSPS) Grant Number 17H04549.

**Acknowledgments:** I would like to thank Masahisa Fujita, Akemi Baba, Tetsuo Mizobe, Katsumi Nakayama, Hideyoshi Sakai, and other referees for their very helpful comments on earlier drafts of the paper.

**Conflicts of Interest:** The author declares no conflict of interest.

## Appendix A. The Model of Fujita and Thisse

This appendix explains the only part of the model of Fujita and Thisse (2003) related to the economies of sequence of this paper. Let $\lambda_r$ be the share of skilled workers in a region r, so that $\lambda_1 \equiv \lambda$ and $\lambda_2 \equiv 1 - \lambda$. Consider the case in which a firm producing a variety of goods can freely decide its location at each time, irrespective of the region in which the goods were innovated. The numbers of modern varieties of goods in region 1 and region 2 are notated as $M_1$ and $M_2$, respectively, and are positive at any given time.

All workers have the same instantaneous utility function, given by

$$u = Q^\mu T^{1-\mu} / \mu^\mu (1 - \mu)^{1-\mu},$$

where T is the consumption of homogeneous traditional goods, $\mu$ is the expenditure share of the modern varieties, and Q stands for the index of the consumption of the modern varieties given by

$$Q = \left[ \int_0^M q(i)^{(\sigma-1)/\sigma} di \right]^{\sigma/(\sigma-1)},$$

where M is the total mass of modern varieties available in the global economy at time t, $\sigma$ is the elasticity of substitution between any two varieties, while q(i) represents the consumption of variety i.

For the chosen value of the share of skilled workers in a region, $\lambda$, $V_r(0;\lambda)$ stands for the lifetime indirect utility of a skilled worker in region r (=A and B) and $v_r(t;\lambda)$ is the corresponding instantaneous indirect utility at time t. This means

$$V_r(0;\lambda) = \int_0^\infty e^{-\gamma t} \ln[v_r(t;\lambda)] dt$$

where $\gamma > 0$ is the subjective discount rate common to all consumers.

Let $E_r$ be the total expenditure in region r at the time in question and $P_r$ be the price index of the modern goods in this region. Then, the total demand for variety i produced in region r equals

$$q_r(i) = \mu E_r p_r(i)P_r + \mu E_s [p_r(i)\Gamma]P_s\Gamma,$$

where r, s = A, B and $r \neq s$, $p_r(i)$ is the price of variety i in region r, and $\Gamma$ accounts for the melting of the variety during its transportation. The corresponding profit is

$$\pi_r(i) = [p_r(i) - 1]q_r(i).$$

The firms' profits must be identical across regions; then, $q_1^* = q_2^*$, where $q_r^*$ denotes the equilibrium output of any variety of goods produced in region r. The equilibrium output of any variety produced in region r and the equilibrium profit are given by

$$q_r^* = \mu\rho[E_r/(M_r + \varphi M_s) + E_s/(M_s + \varphi M_r)], \pi_r^* = q_r^*/(\sigma - 1),$$

where $\varphi \equiv \Gamma^{-(\sigma-1)}$ and $M_r$ represents the number of modern varieties produced in region r at the time in question.

Using $M \equiv M_1 + M_2$ and $E \equiv E_1 + E_2$, we obtain
$$M_1 = (E_1 - \varphi E_2) M/(1 - \varphi) E^*, M_2 = (E_2 - \varphi E_1) M/(1 - \varphi) E^*,$$

so that

$$M_1 > 0, M_2 > 0,$$

If, and only if, $\varphi < E_1/E_2 < 1/\varphi$.

It can be shown that

$$M_1 = M \text{ and } M_2 = 0,$$

If, and only if, $E_1/E_2 \geqq 1/\varphi$.

Next, the paper chooses any $\lambda \in [0,1]$ and the steady-state growth path under that specific $\lambda$. Then,

$$E_1(\lambda)/E_2(\lambda) = [L/2 + \lambda a^*(\lambda)\{\gamma + k_1(\lambda)\}]/[L/2 + (1 - \lambda) a^*(\lambda)\{\gamma + k_2(\lambda)\}],$$

where
$$a^*(\lambda) \equiv M(t)\Pi(t) = \mu E^*/\sigma[\gamma + g(\lambda)], k_1(\lambda) \equiv \lambda[\lambda + \eta(1 - \lambda)]^{1/\beta} \text{ and } k_2(\lambda) \equiv (1 - \lambda)(1 - \lambda + \eta\lambda)^{1/\beta}.$$

It can be verified that

$$E_1(1)/E_2(1) = (\sigma + \mu)/(\sigma - \mu), E_1(1/2)/E_2(1/2) = 1, E_1(0)/E_2(0) = (\sigma - \mu)/(\sigma + \mu).$$
$$d[E_1(\lambda)/E_2(\lambda)]/d\lambda > 0, \lambda \in (0,1).$$

when the transport cost of the modern goods is such that

$$\Gamma^{\Sigma-1} \equiv 1/\varphi \leqq [(\sigma + \mu)/(\sigma - \mu)],$$

and $\lambda$ is larger than $\lambda''$, or $\lambda'' \leqq \lambda$;
then, we have

$$E_1(\lambda)/E_2(\lambda) \geqq 1/\varphi \equiv \Gamma^{\sigma-1},$$

where $\lambda''$ satisfies the equation of $E_1(\lambda'')/E_2(\lambda'') = 1/\varphi$. As shown above,

$$M_1 = M \text{ and } M_2 = 0,$$

The modern goods are agglomerated in region 1, which has a greater share of the innovation sector.

Beijing residency permits contribute to an increase in the share of skilled workers in a region, λ. Under the sufficiently low transport costs below, the policy of raising the share of skilled workers in a region, λ, from λ < λ″ to λ″ ≦ λ, results in $M_1 = M$ and $M_2 = 0$.

When the transport costs of the modern goods, Γ, are low, such that

$$\Gamma \leqq [(\sigma + \mu)/(\sigma - \mu)]^{-(\sigma-1)};$$

then, in the core-periphery structure, the whole innovation sector is agglomerated in the core region.

### Appendix B. Data Sources of Table 6

(1) [PT]: Patents: World Intellectual Property Organization. 2020. Available online: https://www.wipo.int/patentscope/en/data/forms/products.jsp

(2) [VA]: Added value: National Science Foundation. 2020. Available online: https://www.nsf.gov/statistics/

(3) [AC]: Number of Published Science and Engineering Articles: National Science Foundation. 2020. Available online: http://www.nsf.gov/statistics/

(4) [RE]: Research expenses: *OECD Science, Technology and Innovation Outlook 2018*, OECD Publishing, Paris. Available online: https://doi.org/10.1787/sti_in_outlook-2016-en. (accessed on 2 May 2019)

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
