# Peer review of "‘Sequencing Economics’ on the ICT Industry Agglomeration for Economic Integration"

_economies, doi:10.3390/economies9010002_

Round 1
Reviewer 1 Report
The article has introduced the required modifications, improving substantially. Now it is more easily understandable, as well as better justified the union of techniques and methodologies that it proposes. However, I have one last question that I would like the authors to clarify. The proposed methodology fits quite well with the analysis of the Chinese case, where there is a coexistence of market and indicative planning. This coexistence does not occur in other capitalist environments and if it does, it does so on a much smaller scale and with much less power than that observed in the Chinese case (which could possibly be extended to other Asian countries). I think it would be advisable to introduce a reflection on the extent to which the proposed methodology and, specifically, the sequencing of phases proposed in figure 2, can be applied to socio-political and cultural contexts in which the State has the capacity to act on the provision of resources (human, infrastructure...) but limitations in the promotion of business activities.
Author Response
I would like to appreciate the comments by the reviewers of my paper.
They helped me to improve it very much.
Reviewer 1: Comments:
However, I have one last question that I would like the authors to clarify. The proposed methodology fits quite well with the analysis of the Chinese case, where there is a coexistence of market and indicative planning. This coexistence does not occur in other capitalist environments and if it does, it does so on a much smaller scale and with much less power than that observed in the Chinese case (which could possibly be extended to other Asian countries). I think it would be advisable to introduce a reflection on the extent to which the proposed methodology and, specifically, the sequencing of phases proposed in figure 2, can be applied to socio-political and cultural contexts in which the State has the capacity to act on the provision of resources (human, infrastructure...) but limitations in the promotion of business activities.
Response: I agree with this comment.
Six main issues remain to be examined in future. First, this paper used Beijing as a case study for ICT industry agglomeration and obtained tentative conclusions to apply to other cases in other regions around the world. The number of case studies should, however, be increased to ensure the general applicability of conclusions obtained in the paper. Second, the sequencing economics is to be adopted to analyze not only the manufacturing and tourism industry but also other industries despite this paper focusing only on the ZSP ICT industry agglomeration. Third, with regard to statistical analysis, new data should be explored, and new methods of statistical analysis should be developed over and above Granger causality tests. Fourth, the key new segments of an agglomeration need to be identified to clarify substitute data for segments, prove the ‘economies of sequence,’ and establish the ‘sequencing economics’ to agglomeration policy. Fifth, environmental issues related to agglomeration are a key that should be tackled as soon as possible after the Covid-19 pandemic has been controlled. Finally, the feasibility of the proposed methodology is key to completing the building of the segments of an agglomeration. The methodology fits quite well with the analysis of the Chinese case where there is the coexistence of ‘market’ and ‘indicative planning.’ But we need to examine whether the methodology, specifically, the sequencing of phases proposed in Figure 2, can be applied to socio-political and cultural contexts in which the State has the capacity of acting on the provision of resources such as human resources and infrastructure, but limitations on the promotion of business activities.
Reviewer 2 Report
The authors have done very good job in the revised version of the manuscript. I am happy to suggest publication.
Author Response
I would like to appreciate the comments by the reviewers of my paper.
They helped me to improve it very much.
Reviewer 2: Thank you very much for no comment and no response this time.
Reviewer 3 Report
It is a nice piece of research. I strongly recommend the publication in Economies.
Author Response
I would like to appreciate the comments by the reviewers of my paper.
They helped me to improve it very much.
Reviewer 3: Thank you very much for no comment and no response this time.
This manuscript is a resubmission of an earlier submission. The following is a list of the peer review reports and author responses from that submission.
Round 1
Reviewer 1 Report
This is a remarkable article, especially for its ambition, which also deals in depth with an interesting and relevant topic from both a theoretical and an empirical perspective. In this sense, it refers to the successful example of the development of new productive specializations in China. Also, it should be noted that it has a broad and creative theoretical base founded on the dialogue between different interrelated theoretical frameworks
However, independently of these outstanding elements, the article has a series of important deficiencies, motivated in good measure by its own ambitious conception. The main problem is the lack of correspondence between the theoretical argument and the empirical evidence.
“The main two findings are as follows: first, an increase in research expenses Granger-causes an increase in the number of patents; second, an increase in the number of patents Granger-causes an increase in the amount of value added"
In this sense, the sense in which the two main results (cited above) of the research justify or validate the wide range of theoretical statements used to analyze the case study seems questionable.
Another problem to note is that the reference made to the case of the entire OECD does not seem justified either, given the article's emphasis on explaining the case of the Zhongguancun region. In this sense, the article makes no reference to international dynamics or to national differences or specificities. For this reason, the international issue appears to be little connected to the rest of the content of the article on pages 23-25. Given the detail with which the case of the Zhongguancun agglomeration has been presented, it is logical in my opinion to focus the empirical analysis on this case as well.
A last deficiency to be commented on has to do with the theoretical framework used, which is very interesting, but I understand that it implicitly refers to the Chinese case (or in any case, of economies with wide levels of planning and public intervention) and, therefore, with difficulties to be globally generalized. On an international scale, many clusters have been generated spontaneously, without any supporting public policy (see the classical case of the Third Italy), while others have had essentially autonomous dynamics , although they have been able to benefit from certain supporting policies. In this sense, key concepts of this paper such as "lean construction" or "efficiency in construction" have little meaning in autonomously generated clusters beyond policies specifically designed from the public sector.
Author response:
Sequencing economics is used to specify and sequence the segments of an agglomeration from the perspective of ‘economies of sequence.’ The concept of economies of sequence can be defined as the selection of any two segments from among the entire group of segments of an industrial agglomeration and the sequencing of the segments toward the efficient building of the agglomeration. Granger causality tests examine whether the facts of ‘economies of sequence’ are significant or not.
The purpose of this paper is to find the facts of ‘economies of sequence’ in the ICT industry agglomeration, while trying to establish the ‘sequencing economics’ on the agglomeration in architecture theory. The Zhongguancun Science Park in Beijing, China (ZSP) is a case study on the ICT industry agglomeration, which has been taken as a reference. The paper focuses on sequencing economics in architecture theory by taking ‘new economic geography’ into consideration. The model of Fujita and Thisse (2003) derives the starting conditions of the sequencing of the segments of the ICT industry agglomeration, as shown in Figure 1. Integrating the results of the three studies on economies of sequence proposes practical prescriptions for segment sequencing. In short, sequencing economics analyzes the segment building of an ICT agglomeration from the three of the sequencing perspectives of a case study, a model of new economic geography, and Granger causality testing for practical application to a regional development strategy.
These sentences including ‘OECD’ are deleted. “The target of the ASEAN Economic Community 2015 is to enhance infrastructure and communications connectivity. One Belt One Road Initiative is a Chinese project that focuses on improving connectivity. Strengthening physical, intuitional, and people-to-people connectivity promotes regional economic integration by reducing transport costs. Transport costs in new economic geography include the opportunity costs of the days required for transportation, tariffs, non-tariff barriers, and differences in language and cultural customs (Sato et al. (2011)). The reduction of transport costs is a necessary condition for industrial agglomeration, resulting in economic integration. According to Fujita and Thisse (2003), “the main effect of regional integration is likely to be an increase in economic efficiency within the spatial economy”. Herein, we focus on regional integration from an agglomeration perspective.”
These sentences including "lean construction" or "efficiency in construction"are deleted.
Kuchiki (2020a) and Kuchiki (2020b) illustrated industrial agglomeration policies succeeded in Asia.
The following sentences are inserted in the introduction.
With regard to sequencing economics, Kuchiki (2020a) analyzed manufacturing industry agglomerations, including the electric/electronics industry and the automobile industry, as shown in Figure 2. It showed successful agglomeration policies in East Asia from the 1980s. Kuchiki (2020b) also applied the sequencing economics to tourism industry agglomerations.
With regard to new economic geography, Krugman (1991) proposed a fundamental model by introducing the mobility of skilled workers. Martin and Ottaviano (1999) developed a dynamic version of this fundamental model by imposing an assumption that research and development is the source of growth. Hirose (2008) analyzed the case of migration and agglomeration using a dynamic version. Fujita and Thisse (2003) assumed that skilled workers are mobile and that patents for new products can be transferred without cost between regions. As shown in Figure 1, their model can be applied to a model for a two-region conglomeration across two counties to drive the conditions of an ICT agglomeration. With regard to sequencing economics, Kuchiki (2020a) analyzed manufacturing industry agglomerations, including the electric/electronics industry and the automobile industry, as shown in Figure 2. It showed successful agglomeration policies in East Asia from 1980s. Kuchiki (2020b) also applied the sequencing economics to tourism industry agglomerations. With regard to ICT agglomerations, Han, Liu, Rosa, and Li (2019) and Monti and Ponci (2017) published special issues on urban agglomeration and ICT, respectively. Chen, Huang, Liu, Luan, and Song (2019) and Han, Zhang, Cai, and Ma (2019) also analyzed high-tech industry agglomerations based on green economy efficiency and industrial agglomeration based on land use efficiency, respectively. However, sequencing economics on the ICT industry agglomeration is yet to be analyzed. Table 2 shows the segments of an ICT industry, including researchers and funding. The building of the segments of an agglomeration can be started once they are sequenced efficiently. The economics seeks to reveal the ‘economies of sequence’; i.e., how the efficient construction of the segments of an agglomeration should be sequenced. Thus, this paper applies the model of Fujita and Thisse (2003) to the case of a two-region model across two countries to drive the starting conditions for constructing agglomerations in the ICT industry.
Reviewer 2 Report
I am negative for this paper for the following 4 major reasons.
- The paper is not clear and well-written. It is very confusing using a mix of arguments without a clear connection.
- In the introduction there could also be a clearer statement on the relevance of the present paper, what is the justification of the present paper; what is the research question of the paper? How can the paper contribute to the literature?
- The paper uses many theories but there is no any connection between them. It lacks conceptual/theoretical clarity. I found the arguments to be under motivated for a full-fledged research article.
- There is no attempt to articulate a research question that would aim to shed light on an issue of general interest of agglomeration and no sound theoretical grounding of the discussion of the empirical evidence presented. The analysis is not guided by the attempt to address a clear research question that goes beyond the aim to provide a purely descriptive account of the phenomenon of agglomeration.
Hence, I reject the manuscript.
